# Multi-Object 3D Grounding with Dynamic Modules and Language-Informed Spatial Attention

**Haomeng Zhang    Chiao-An Yang    Raymond A. Yeh**
Department of Computer Science, Purdue University
`{zhan5050, yang2300, rayyeh}@purdue.edu`

## Abstract

Multi-object 3D Grounding involves locating 3D boxes based on a given query phrase from a point cloud. It is a challenging and significant task with numerous applications in visual understanding, human-computer interaction, and robotics. To tackle this challenge, we introduce D-LISA, a two-stage approach incorporating three innovations. First, a dynamic vision module that enables a variable and learnable number of box proposals. Second, a dynamic camera positioning that extracts features for each proposal. Third, a language-informed spatial attention module that better reasons over the proposals to output the final prediction. Empirically, experiments show that our method outperforms the state-of-the-art methods on multi-object 3D grounding by 12.8% (absolute) and is competitive in single-object 3D grounding.[1]

## 1 Introduction

Building agents that can operate in real-world environments with humans has been a fundamental goal of artificial intelligence. Importantly, the agent would need to understand the 3D scene and natural language to take instructions from humans. To benchmark these capabilities, there is an increasing amount of interest in the task of object grounding in 3D [2, 4, 8, 17, 21, 23, 39, 40, 46, 50]. Recently, the task of multi-object 3D grounding [52] has been proposed, *i.e.*, given a text description and a 3D scene localize *all* objects referred by the description.

Along with the benchmark, Zhang et al. [52] proposes, M3DRef-CLIP, a two-stage approach that first detects all the potential objects (capped at a maximum number) from the 3D scene, and then reasons about which of the objects are relevant to the text description by extracting features for each of the objects. Specifically, they leverage both 3D features from the point cloud, and 2D features extracted from renderings of the detected objects at fixed camera poses. These object features along with the text embedding are passed into a Transformer to make the final prediction. Model training is formulated as multi-output classification, where each potential object is classified based on whether it is referred to by the text.

In this work, we identify several directions in which M3DRef-CLIP could be improved. First, the generation of object proposals is based on a *fixed* maximum. Prior work [30] points out the dilemma of deciding the number of boxes in the 3D grounding task under the two-stage detection-and-selection diagram. Excessive proposals may increase complexity and lead to redundant computations while sparse proposals may miss critical information in the scene. Second, the camera poses of the renderer are *fixed* to hand-selected viewpoints, which seems unlikely to be ideal given the variability in object sizes. Third, the fusion module does not effectively reason over the spatial relationship of the objects based on the text description.

---

[1]Project page: `https://haomengz.github.io/dlisa`
 Code: `https://github.com/haomengz/D-LISA`

38th Conference on Neural Information Processing Systems (NeurIPS 2024).

To address these shortcomings, we propose D-LISA, a two-stage approach that incorporates three innovative modules. First, instead of using all detected objects, we use a dynamic proposal module to select the key box proposals. Second, we incorporate a dynamic multi-view renderer module that optimizes the viewing angles tailored to a specific scene. Third, we introduce a language-informed spatial fusion module that uses textual description to guide reasoning based on spatial relations.

To evaluate our proposed method, we conduct experiments on the Multi3DRefer benchmark for multi-object 3D grounding and achieve a substantial 12.8% absolute increase over the existing baseline M3DRef-CLIP. We also validate the effectiveness of our method by achieving the state-of-the-art performance on ScanRefer benchmark [8] and competitive results on Nr3D benchmark [2] for single-object 3D grounding. **Our contributions are summarized as follows:**

- We introduce a dynamic box proposal module that automatically determines the key box proposals for the later reasoning stage, which could potentially replace the fixed object proposals prevalent in existing two-stage grounding pipelines. Also, we learn the camera pose for 2D rendering dynamically based on the scene, enhancing the quality of auxiliary object features in uncertain environments.

- We propose a language-informed spatial fusion module that dynamically captures the spatial relations among objects, significantly improving the model's contextual understanding and performance in the multi-object 3D grounding task.

- We conduct thorough experiments to validate the proposed framework. The proposed approach not only significantly outperforms the state-of-the-art model in multi-object 3D grounding, but also maintains robust performance in the single-object 3D grounding task.

## 2 Related Work

**2D grounding** aims to identify the target object in a 2D image based on a natural language description. The conventional detection-and-selection two-stage pipeline first extracts the visual features for the proposals and language features for the description then employs the attention mechanism to effectively align the visual features and language features [15, 27, 42, 48, 55]. Alternatively, one-stage methods directly regress the target boxes by integrating object detection and language understanding [28, 33, 44, 45]. While relational graphs have been used to explicitly model the object relations in 2D images [29, 37, 43], extending the modeling to 3D is challenging due to larger number of objects and more complex spatial relations.

**3D grounding.** Similar to 2D grounding, 3D grounding aims to target the language-referred object in a 3D scene. There have been a variety of datasets [1, 2, 8] and approaches [5, 17, 39, 40, 50] to tackle this challenging problem. M3DRef-CLIP [52] is the pioneered work to explore targeting multiple objects that match the language description. Other than the one-stage methods that directly identify the target box [30, 38], two-stage methods like M3DRef-CLIP following the detection-and-selection diagram are facing the issue of determining the number of boxes from the detection stage. We propose a module that dynamically selects the key box proposals from object candidates.

2D features have been widely used to assist with 3D grounding [4, 17, 21, 23, 46] as well as other 3D tasks [3, 22, 31, 36, 47, 51]. However, most studies rely on fixed camera poses to generate these 2D image features, which is sub-optimal given the varying object sizes across different 3D scenes. In contrast, we propose to learn scene-conditioned camera poses for object rendering.

Many works have studied how to model the object relations in complex 3D scenes [7, 16, 18–20, 34, 49, 53]. For example, 3DVG-Trans [53] and M3DRef-CLIP [52] model the spatial relations based on distances. ViL3DRef [9] and CORE-3DVG [41] incorporate language and hand-selected features to guide the spatial relations. Differently, we propose a simple yet effective language-informed balancing strategy to explicitly reason over the spatial relation that solely depends on distances.

## 3 Approach

Given a 3D point cloud of a scene $\mathcal{S}$, and a text description $\mathcal{T}$, the task of multi-object 3D grounding aims to predict the set of bounding boxes $\mathcal{P}$ for objects that are referred to in the text description. Our proposed Multi-Object 3D Grounding with **D**ynamic Modules and **L**anguage **I**nformed **S**patial

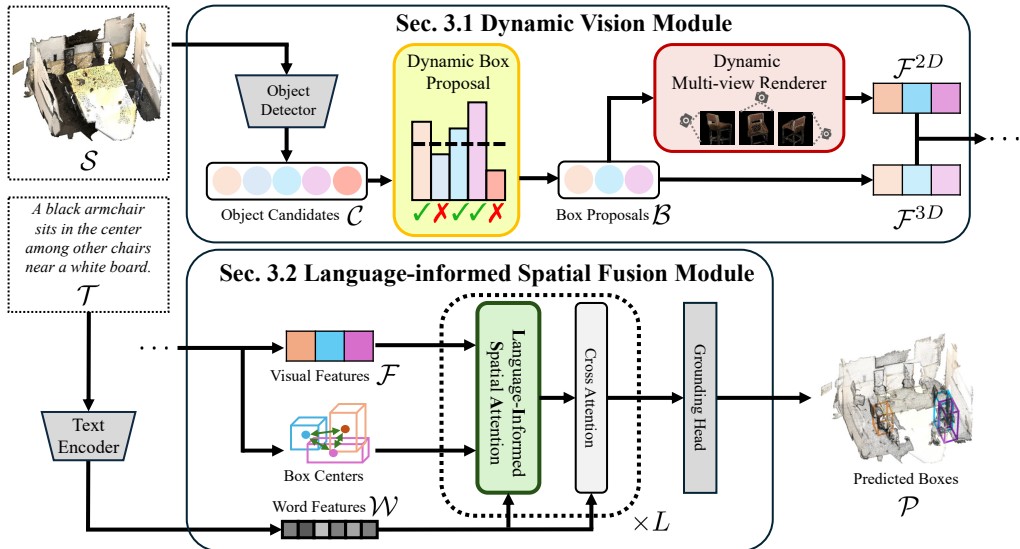

Figure 1: **Illustration of the overall pipeline.** Our D-LISA processes the 3D point cloud through the dynamic visual module (Sec. 3.1) and encodes the text description through a text encoder. The visual and word features are fused through a language informed spatial fusion module (Sec. 3.2).

**A**ttention (**D-LISA**) follows the detection-and-selection paradigm for multi-object 3D grounding task [52]. This paradigm involves three components: (i) a text encoder to extract text features; (ii) a vision module to detect object proposals and extract corresponding features given a point cloud; (iii) a fusion module that combines the text and object features to select the final referred bounding-boxes. Specifically, our D-LISA is designed with a novel vision module that allows for a dynamic number of proposal boxes and extracts features from dynamic viewpoints (Sec. 3.1) per scene. Furthermore, we propose a fusion model that is spatially aware with explicit language conditioning (Sec. 3.2). An overview of our approach is illustrated in Fig. 1.

## 3.1 Dynamic Vision Module

Our dynamic vision module takes a 3D scene point cloud $\mathcal{S}$ as the input and generates a set of box proposals $\mathcal{B}$ with corresponding visual features $\mathcal{F}$. As in prior work [52], we adopt the backbone detector of PointGroup [25] to obtain a fixed number of $M$ box candidates $\mathcal{C}$, *i.e.*, $|\mathcal{C}| = M$. To eliminate irrelevant detected objects, we employ a dynamic box proposal module with non-maximum suppression (NMS). This module dynamically selects a subset with variable sizes, from the $M$ candidates, to form the set of box proposals $\mathcal{B}$, which are then used by the fusion model.

**Dynamic box proposal.** To achieve box proposals with a flexible number, we learn a dynamic proposal probability $\alpha_m$ for each of the $M$ box candidates.

We model the proposal probability $\alpha_m$ as a normalized linear function of the detector score $s_m$, *i.e.*,

$$\alpha_m = \text{Sigmoid}(\text{Linear}(s_m)). \tag{1}$$

At prediction time, an object candidate would be selected if the dynamic proposal probability exceeds the filtering threshold $\tau_f$:

$$\mathcal{B}' = \{b_m \in \mathcal{C} \mid \alpha_m > \tau_f\}, \tag{2}$$

where $b_m$ denotes the 3D box of the $m^{\text{th}}$ object.

We then use non-maximum suppression (NMS) [14] to remove overlapping boxes from the box proposal candidates $\mathcal{B}'$ and finalize the box proposals $\mathcal{B}$. First, the proposal probabilities $\alpha_i$ are sorted in descending order. Then we sequentially select the candidate with the highest probability as a box proposal and remove other box proposal candidates that have an Intersection over Union (IoU) greater than a threshold $\tau_{\text{NMS}}$. The NMS module ensures the box proposals $\mathcal{B}$ do not include duplicated boxes for the same object.

*Dynamic Proposal loss.* To train this proposal probability, we incorporated a regularization term penalizing the expected value of the number of boxes

$$\mathcal{L}_{\text{dyn}} = \sum_{m=1}^{M} \alpha_m. \tag{3}$$

This loss encourages the model to use as few box proposals as possible while maintaining the grounding performance.

**Object proposal feature extraction.** Given the $N$ box proposals $\mathcal{B}$, *i.e.*, $|\mathcal{B}| = N$, we extract visual features $\mathcal{F}$ that is a concatenation of both the 3D features $\mathcal{F}^{\text{3D}}$ from the detector and 2D features $\mathcal{F}^{\text{2D}}$ from our dynamic multi-view renderer.

*3D feature from detector backbone.* Each box $b_n$ in the box proposals $\mathcal{B}$ has a corresponding 3D feature $\boldsymbol{f}_i^{\text{3D}}$ that can be extracted from the detector backbone. Next, to ensure that the proposal probability $\alpha_m$ reflect the quality of the box $b_m$, we weight the 3D features with the probability, *i.e.*,

$$\mathcal{F}^{\text{3D}} = \{\alpha_1 \cdot \boldsymbol{f}_1^{\text{3D}}, \alpha_2 \cdot \boldsymbol{f}_2^{\text{3D}}, \ldots, \alpha_N \cdot \boldsymbol{f}_N^{\text{3D}}\}. \tag{4}$$

*2D feature from Dynamic multi-view renderer.* The dynamic multi-view renderer takes as input the box proposals $\mathcal{B}$ and generates the corresponding 2D features $\mathcal{F}^{\text{2D}}$. Instead of using fixed camera poses for rendering all objects across different scenes, we learn scene-conditioned camera poses for rendering. We predefined $V$ base camera poses $\boldsymbol{d}_j^{\text{cam}}$ for $j = 1, 2, \ldots, V$. Next, we calculate the average size of all boxes denoted as $\bar{\boldsymbol{q}} \in \mathbb{R}^3$ with the average length, width, and height respectively. We use a Multi-Layer Perceptron (MLP) to learn the camera pose offset for each view $j$ based on the average box size $\bar{\boldsymbol{q}}$:

$$\Delta \boldsymbol{p}_j^{\text{cam}} = \text{MLP}_j(\bar{\boldsymbol{q}}). \tag{5}$$

The final camera pose for each view $j$ is

$$\boldsymbol{p}_j^{\text{cam}} = \boldsymbol{d}_j^{\text{cam}} + \Delta \boldsymbol{p}_j^{\text{cam}}. \tag{6}$$

For each view $j$, the renderer generates the 2D image for each box proposal $b_i$ with camera pose $\boldsymbol{p}_j^{cam}$. The pre-trained CLIP image encoder extracts the 2D features for each view. Finally, we compute the average over all the extracted features from each view to obtain the 2D features

$$\mathcal{F}^{\text{2D}} = \left\{ \frac{1}{V} \sum_{j=1}^{V} \text{CLIP}(\text{Render}(b_n, \boldsymbol{p}_j^{\text{cam}})) \,\middle|\, b_n \in \mathcal{B} \right\}. \tag{7}$$

### 3.2 Language-Informed Spatial Fusion Module

Given the visual features $\mathcal{F}$ from the dynamic vision module and the word features $\mathcal{W}$ from CLIP's text encoder, the language-informed spatial fusion module predicts a probability $p_n$ on whether the object in box $b_n$ is targeted in the text description. The module consists of a stack of transformer layers followed by an MLP grounding head.

To better capture the spatial relationship among objects, we introduce the language-informed spatial attention (LISA) block that balances the visual attention weights and the spatial relations using the sentence feature $\boldsymbol{g}$, a weighted sum over all word features. Each transformer layer comprises a language-informed spatial attention block and a cross-attention block, as illustrated in Fig. 1. Finally, we only predicted a box if the associated probability $p_n$ exceeds a threshold $\tau_{\text{pred}}$, *i.e.*, the predicted box set is

$$\mathcal{P} = \{b_n \mid p_n > \tau_{\text{pred}}\}. \tag{8}$$

We now discuss the details of LISA. The details of the cross-attention block are provided in Appendix Sec. A4.

**Language informed spatial attention (LISA).** Given the visual feature matrix $\boldsymbol{F} = [\boldsymbol{f}_1, \boldsymbol{f}_2, \ldots, \boldsymbol{f}_N]^T \in \mathbb{R}^{N \times d_o}$ where $\boldsymbol{f}_n \in \mathcal{F}$ and the sentence feature $\boldsymbol{g}$, language-informed spatial attention block (Fig. 2) updates the visual features with spatial information by balancing the visual attention weights and spatial relations guided by language.

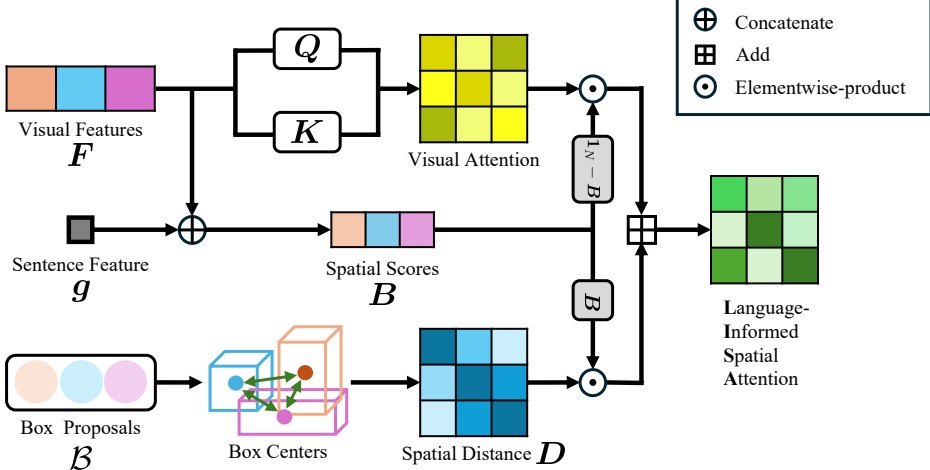

Figure 2: **Illustration of language informed spatial attention (LISA).** We model the object relations through spatial distance $D$. For each box proposal, a spatial score is predicted to balance the visual attention weights and spatial relations.

LISA follows the standard self-attention mechanism proposed by Vaswani et al. [35] consisting of queries, keys, and values. Given $F$, the queries $Q$, keys $K$ and values $V$ are computed as follows:

$$Q = FW_Q, \quad K = FW_K, \quad V = FW_V \tag{9}$$

with linear projections $W_{Q/V/K} \in \mathbb{R}^{d_o \times d}$.

To explicitly build in spatial reasoning, we introduce spatial scores $B$, conditioned on the sentence feature and visual features, to weight between the standard attention terms and a spatial distance matrix $D$. The overall language-informed spatial attention is as follows:

$$\text{LISA}(F, g, D) = \text{softmax}\left((\mathbf{1}_N - B) \odot \frac{QK^T}{\sqrt{d}} + B \odot D\right) V, \tag{10}$$

where $\mathbf{1}_N$ is an all-ones matrix and $\text{softmax}$ normalizes along each row. We now describe $B$ and $D$.

*Spatial scores $B$.* Given a variety of objects in a complex scene, we want the model to dynamically learn whether an object should pay more attention to the spatial relationship based on text description. For the $i^{\text{th}}$ object in the box proposal, we predict the normalized score $\beta_i$ by concatenating the visual feature $f_i$ and the sentence feature $g$, followed by a linear projection. To align with the attention weights, we construct the spatial scores $B \in \mathbb{R}^{N \times N}$ as

$$B = \begin{bmatrix} \beta_1 & \beta_1 & \cdots & \beta_1 \\ \beta_2 & \beta_2 & \cdots & \beta_2 \\ \vdots & \vdots & \ddots & \vdots \\ \beta_N & \beta_N & \cdots & \beta_N \end{bmatrix}, \quad \text{where } \beta_i = \text{Sigmoid}(\text{Linear}(g \oplus f_i)) \tag{11}$$

and $\oplus$ denotes a concatenation.

*Spatial distance matrix $D$.* We model the spatial relationship among objects through relative distances. We construct this matrix $D \in \mathbb{R}^{N \times N}$ by computing the pairwise $l_2$-distance between the box centers $c_i$ and $c_j$, *i.e.*, $d_{ij} = ||c_i - c_j||_2$. To ensure that closer objects should receive greater attention, we define $D_{ij} = \frac{1}{d_{ij}}$.

### 3.3 Training details

In addition to the dynamic proposal loss for our dynamic box proposal module, we follow the loss functions of the prior work [52] for end-to-end training. These include detection loss, reference loss, and contrastive loss. We briefly discuss these losses for completeness.

Table 1: Quantitative comparison of F1@0.5 on the Multi3DRefer [52] val set.

| Method | F1@0.5 on Val (↑) | | | | | |
|---|---|---|---|---|---|---|
| | ZT w/o D | ZT w/D | ST w/o D | ST w/D | MT | All |
| 3DVG-Trans [53] | 87.1 | 45.8 | 27.5 | 16.7 | 26.5 | 25.5 |
| D3Net [9] | 81.6 | 32.5 | 38.6 | 23.3 | 35.0 | 32.2 |
| 3DJCG [6] | **94.1** | **66.9** | 26.0 | 16.7 | 26.2 | 26.6 |
| M3DRef-CLIP [52] | 81.8 | 39.4 | 47.8 | 30.6 | 37.9 | 38.4 |
| M3DRef-CLIP w/NMS | 79.0 | 40.5 | **67.6** | 40.0 | 49.1 | 49.3 |
| D-LISA | 82.4 | 43.7 | 67.1 | **42.5** | **51.0** | **51.2** |

*Detection loss.* We use Pointgroup [25] as our detector backbone and adopt their training losses. The detection loss $\mathcal{L}_{\text{det}}$ consists of four components: a) a semantic segmentation loss, b) an offset regression loss, c) an offset direction loss, and d) a proposal score loss.

*Reference loss.* For multi-object 3D grounding, we adopt the binary cross-entropy loss over the detected objects as the reference loss $\mathcal{L}_{\text{ref}}$. We apply the Hungarian algorithm [26] to find an optimal match based on the pairwise IoU between the detected objects and ground truth. A detected box is successfully grounded if it matches one ground truth box in the Hungarian solution and the pairwise IoU is greater than a threshold $\tau_{\text{train}}$. For single-object 3D grounding, we use the cross-entropy loss. We identify the highest IoU between the detected boxes and the ground truth box and consider it a success if this maximal IoU is greater than the threshold $\tau_{\text{train}}$.

*Contrastive loss.* We apply a symmetric contrastive loss $\mathcal{L}_{\text{ctr}}$ between the object features and the word features. A positive pair is formed if the object features and the word features come from the same scene-instruction pair, while a negative pair is formed if they come from different scene-instruction pairs. For computing efficiency, we only identify the positive and negative pairs within a single batch. This loss has been proven effective for learning better multi-modal embeddings [52].

The total loss function is a weighted sum over all loss terms

$$\mathcal{L} = \lambda_{\text{det}}\mathcal{L}_{\text{det}} + \lambda_{\text{ref}}\mathcal{L}_{\text{ref}} + \lambda_{\text{ctr}}\mathcal{L}_{\text{ctr}} + \lambda_{\text{dyn}}\mathcal{L}_{\text{dyn}}, \tag{12}$$

where $\lambda_i$ is the individual loss weight for each loss term $\mathcal{L}_i$.

## 4 Experiments

We conduct experiments on the Multi3DRefer [52] dataset. We also compare our model with other two-stage methods on single-object grounding using the ScanRefer [8] and the Nr3D [2] datasets. Finally, we ablate the effectiveness of each proposed module.

### 4.1 Multi-object 3D grounding

**Dataset and evaluation metric.** Multi3DRefer is a dataset based on ScanRefer [8]. It contains 61,926 descriptions of 11,609 objects, with each text description potentially referencing zero, single, or multiple target objects.

Using the standard evaluation protocol [52], we report the F1 score at the intersection over union (IoU) threshold of 0.5 over five different categories: a) zero target without distractors of the same semantic class (ZT w/o D); b) zero target with distractors (ZT w/D); c) single target without distractors (ST w/o D); d) single target with distractors (ST w/D); and e) multiple targets (MT). The average over these categories is reported as an overall score.

**Baselines.** Following prior work [52], we consider two-stage methods that perform well on the ScanRefer dataset as baselines; including, 3DVG-Trans [53], D3Net [9], 3DJCG [6] and M3DRef-CLIP [52]. We also report the performance of M3DRef-CLIP with NMS after the first-stage detector for a fair comparison.

**Implementation details.** We train our model on a single NVIDIA A100 GPU. We set the batch size to 4 with the AdamW optimizer using a learning rate of $5e^{-4}$. We follow the same train/val set split as the baselines [52]. For the PointGroup detector, we use the same pre-trained PointGroup

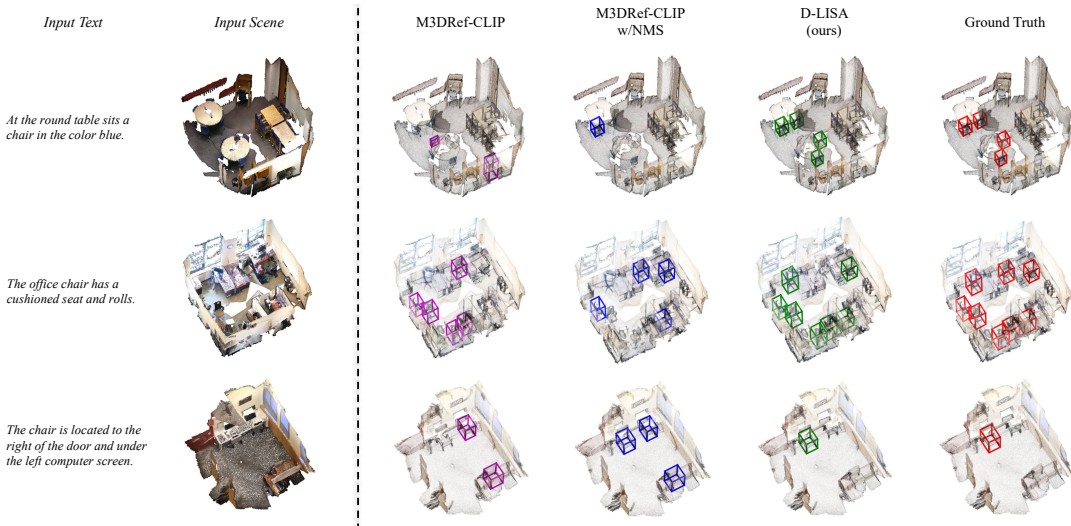

| Input Text | Input Scene | M3DRef-CLIP | M3DRef-CLIP w/NMS | D-LISA (ours) | Ground Truth |

*At the round table sits a chair in the color blue.*

*The office chair has a cushioned seat and rolls.*

*The chair is located to the right of the door and under the left computer screen.*

Figure 3: **Qualitative examples of Multi3DRefer val set.** For each scene-text pair, we visualize the predictions of M3DRef-CLIP, M3DRef-CLIP w/NMS, D-LISA and ground truth labels in magenta/blue/green/red separately.

module following Zhang et al. [52] with the same loss coefficients. We set the dynamic proposal loss coefficient $\alpha_{dyn}$ to 5. We set the $\tau_{\text{train}}$ to 0.25 and search for the optimal value of $\tau_{\text{pred}}$ over {0.05, 0.1, 0.15, 0.2, 0.25} during evaluation for M3DRef-CLIP w/NMS and our model.

**Results.** We compare the F1@0.5 metric of our model and state-of-the-art baselines on Multi3DRefer val set in Tab. 1. Our D-LISA achieves a 12.8% absolute increase in the overall F1@0.5 score over M3DRef-CLIP. Comparing M3DRef-CLIP and M3DRef-CLIP w/NMS, we observe that NMS is a key factor in the final F1 score, successfully removing duplicate predictions leading to improved recall.

Next, D-LISA achieves a better overall F1 score, especially for multiple targets and sub-categories where the distractors of the same semantic class exist. We further provide qualitative results over our method and the baselines in Fig. 3. The top two rows are examples from multiple target categories. Our D-LISA successfully identifies more objects that match the text description. The last row shows an example of a single target with distractors. Our D-LISA accurately identifies the object while the baselines are affected by the distractors and predict additional incorrect targets.

### 4.2 Single-object 3D grounding.

**Dataset and evaluation metric.** We evaluate the single-object 3D grounding performance on the ScanRefer and the Nr3D datasets. The ScanRefer dataset contains 51,583 human-written sentences for 800 scenes in ScanNet [13]. ScanRefer divides scenes into "Unique" and "Multiple" subsets based on whether the semantic class of the target object is unique in the scene.

The Nr3D dataset consists of 41,503 human-annotated text descriptions across 707 indoor scenes from ScanNet. Nr3D divides scenes into "Easy" and "Hard" subsets based on whether there exist the distractors of the same semantic class, and into "View-dependent" and "View-independent" subsets based on whether a specific viewpoint is required to identify the target. Both ScanRefer and Nr3D are annotated for single-object grounding. Different from ScanRefer, Nr3D *assumes perfect object proposals* are provided.

Following prior work [52], for the ScanRefer dataset we report Acc@0.5 on both val and test sets over different subsets. The number represents the proportion of predicted target boxes that have an IoU value greater than 0.5 compared to the ground truth box. For the Nr3D dataset, we report the accuracy of selecting the target bounding box among all candidate proposals on the test set over different subsets.

Table 2: Acc@0.5 of different methods on the ScanRefer dataset [8]. For joint models indicated by *, the best grounding performance with extra captioning training data is reported.

| Method | Acc@0.5 on Val (↑) | | | Acc@0.5 on Test (↑) | | |
|---|---|---|---|---|---|---|
| | Unique | Multiple | All | Unique | Multiple | All |
| TGNN [20] | 56.8 | 23.2 | 29.7 | 58.9 | 25.3 | 32.8 |
| FFL-3DOG [16] | 67.9 | 25.7 | 34.0 | - | - | - |
| InstanceRefer [49] | 66.8 | 24.8 | 32.9 | 66.7 | 26.9 | 35.8 |
| 3DVG-Trans [53] | 62.0 | 30.3 | 36.4 | 57.9 | 31.0 | 37.0 |
| 3DJCG* [6] | 64.3 | 30.8 | 37.3 | 60.6 | 31.2 | 37.8 |
| D3Net* [9] | 72.0 | 30.1 | 37.9 | 68.4 | 30.7 | 39.2 |
| UniT3D* [12] | 73.1 | 31.1 | 39.1 | - | - | - |
| HAM [10] | 67.9 | 34.0 | 40.6 | 63.7 | 33.2 | 40.1 |
| CORE-3DVG [41] | 67.1 | 39.8 | 43.8 | - | - | - |
| M3DRef-CLIP [52] | **77.2** | 36.8 | 44.7 | **70.9** | 38.1 | 45.5 |
| M3DRef-CLIP w/NMS | 75.6 | 38.5 | 45.7 | - | - | - |
| D-LISA | 75.5 | **40.0** | **46.9** | 69.0 | **39.7** | **46.3** |

Table 3: Grounding accuracy of different methods on Nr3D dataset [2].

| Method | Accuracy on Test (↑) | | | | |
|---|---|---|---|---|---|
| | Easy | Hard | View-Dep | View-Indep | All |
| TGNN [20] | 44.2 | 30.6 | 35.8 | 38.0 | 37.3 |
| InstanceRefer [49] | 46.0 | 31.8 | 34.5 | 41.9 | 38.8 |
| 3DVG-Trans [53] | 48.5 | 34.8 | 34.8 | 43.7 | 40.8 |
| FFL-3DOG [16] | 48.2 | 35.0 | 37.1 | 44.7 | 41.7 |
| HAM [10] | 54.3 | 41.9 | 41.5 | 51.4 | 48.2 |
| M3DRef-CLIP [52] | 55.6 | 43.4 | 42.3 | 52.9 | 49.4 |
| D-LISA | **60.2** | **46.2** | **44.3** | **57.4** | **53.1** |

**Baselines.** We focus on comparing the two-stage methods designed for the situation where the ground truth box proposals are not provided. For the ScanRefer dataset, we compare with the baselines: TGNN [20], FFL-3DOG [16], InstanceRefer [49], 3DVG-Trans [53], 3DJCG [6], D3Net [9], UniT3D [12], HAM [10], CORE-3DVG [41] and M3DRef-CLIP [52]. For joint captioning and grounding models 3DJCG, D3Net, and UniT3D, we compare their best grounding performance with extra captioning training data. For the Nr3D dataset, we compare with the above baselines which reported the performance in their paper.

**Implementation details.** We follow the multi-object setting to adapt to the single-object setting. Differently, we let the fusion module return the *most likely* box among all the proposal boxes instead of using a threshold. For the Nr3D dataset, we follow the prior work [52] to directly crop the box features from the detector backbone based on the ground truth bounding boxes. We follow the same train/val/test set split for both datasets as the baselines.

**Results.** We report the Acc@0.5 of different methods on the ScanRefer val set and test set in Tab. 2. Comparing M3DRef-CLIP and M3DRef-CLIP w/NMS, we could see that non-maximum suppression slightly improves the performance. Our D-LISA outperforms all existing baselines on both the ScanRefer val set and test set, especially for the subsets where there are multiple objects with the semantic class of the target object in the scene.

Next, we report the grounding accuracy of different methods on the Nr3D test set in Tab. 3. Our D-LISA outperforms all baselines on the Nr3D test set over all subsets. For more comparison with other methods on the ScanRefer and the Nr3D datasets, see Sec. A2 in the Appendix.

*Limitations:* As with other two-stage methods, the grounding performance of our designed two-stage model is upper bounded by the detector quality. From Tab. 1 and Tab. 2, we can see that our model achieves better performance for complex scenarios but sacrifice some performance for the simpler single-object settings.

Table 4: Ablation study of proposed modules on Multi3DRefer dataset. 'LIS.', 'DBP.' and 'DMR.' stands for 'Language informed spatial fusion', 'Dynamic box proposal', and 'Dynamic multi-view renderer' respectively.

| Row # | LIS. | DBP. | DMR. | F1@0.5 on Val (↑) | | | | | |
|---|---|---|---|---|---|---|---|---|---|
| | | | | ZT w/o D | ZT w/D | ST w/o D | ST w/D | MT | All |
| 1 | ✗ | ✗ | ✗ | 79.0 | 40.5 | **67.6** | 40.0 | 49.1 | 49.3 |
| 2 | ✗ | ✗ | ✓ | 79.5 | 42.3 | 66.1 | 41.2 | 49.2 | 49.8 |
| 3 | ✗ | ✓ | ✗ | 80.3 | 41.5 | 66.2 | 41.4 | 50.6 | 50.3 |
| 4 | ✓ | ✗ | ✗ | 80.1 | 42.6 | 66.6 | 41.8 | 49.9 | 50.4 |
| 5 | ✓ | ✓ | ✓ | **82.4** | **43.7** | 67.1 | **42.5** | **51.0** | **51.2** |

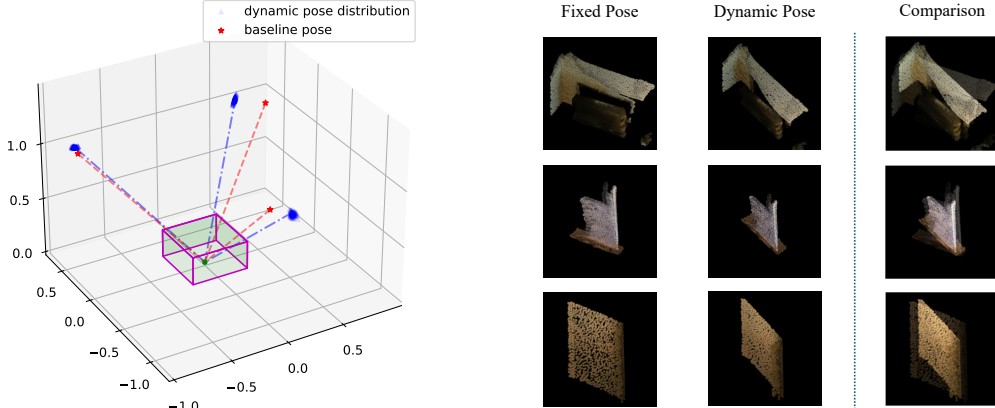

(a) Dynamic pose distribution and fixed baseline pose on Multi3DRefer val set.

(b) Examples of rendered 2D images through dynamic camera pose *vs.* fixed camera pose.

Figure 4: **Qualitative results of dynamic multi-view renderer.** On the left, we show the learned pose distribution over the Multi3DRefer val set and visualize one camera ray example. On the right, we present examples of comparison between rendering with fixed pose and dynamic learned pose.

## 4.3 Ablation studies

We conduct ablation studies on the proposed modules to validate their effectiveness under the multi-object grounding setting on the M3DRef dataset. The ablations follow the same experiment settings for the multi-object grounding in Sec. 4.1. The baseline Row #1 shows the result of M3DRef-CLIP w/NMS.

**Dynamic box proposal.** In Tab. 4, comparing Row #3 with baseline Row #1, we validate the effectiveness of the dynamic box proposal module. We also validate the number of box candidates in the reasoning stage after using the dynamic box proposal module. For our complete model Row #5, an average of 30.5 boxes are selected for the fusion stage on the M3DRef val set. This is a much smaller number of boxes compared to the 62.4 boxes used in baseline Row #1.

**Dynamic multi-view renderer.** In Tab. 4, comparing Row #2 with baseline Row #1, we validate the effectiveness of the dynamic multi-view renderer module. We provide the qualitative results for the dynamic multi-view renderer in Fig. 4. Instead of using fixed camera poses, the dynamic renderer adapts different camera poses from scene to scene, enhancing the quality of 2D object features.

**Language informed spatial fusion.** In Tab. 4, comparing Row #4 with baseline Row #1, we validate the effectiveness of the language-informed spatial fusion module, especially for the sub-categories where distractors exist (ZT w/D and ST w/D). For more ablation results on the language-informed spatial fusion module, please refer to Appendix Sec. A3.

**Computational cost.** We report the FLOPs and inference time of each proposed module and a comparison with the baseline model M3DRef-CLIP in Tab. 5. All experiments are conducted on Multi3DRefer validation set on a single NVIDIA A100 GPU. The reported FLOPs and inference time are the average over the validation set. We observe that the dynamic box proposal module

Table 5: Computational cost for proposed modules during inference.

| Module | FLOPs | Inference time |
|---|---|---|
| Baseline detector | 943.1 M | 0.235 s |
| Detector w/ dynamic box proposal | 943.1 M | 0.241 s |
| Baseline multi-view renderer | 638.9 G | 0.271 s |
| Dynamic multi-view renderer | 638.9 G | 0.276 s |
| Baseline fusion | 155.3 M | 0.004 s |
| Language-informed spatial fusion | 247.4 M | 0.007 s |

and the dynamic multi-view renderer in the dynamic vision module contribute marginally to the computation. The additional computations in the language-informed spatial fusion module are also minimal. In other words, our model achieves better grounding performance without significantly increasing computations.

## 5 Conclusion

In this paper, we present D-LISA, a two-stage pipeline for multi-object 3D grounding, featuring three novel components. Our dynamic box proposal module dynamically selects the key box proposals from detected objects. We enhance the 2D features through optimized scene-conditioned rendering poses using a dynamic multi-view renderer. Furthermore, our language-informed spatial fusion module facilitates explicit reasoning over the object spatial relations. Our proposed approach not only outperforms the state-of-the-art model in multi-object 3D grounding but also is competitive in single-object 3D grounding.

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

## Appendix

The appendix is organized as follows:

- In Sec. A1, we provide additional results on the Multi3DRefer dataset for multi-object grounding.
- In Sec. A2, we provide additional comparisons with state-of-the-art methods on ScanRefer and Nr3D datasets for single-object grounding.
- In Sec. A3, we provide additional comparisons and ablation results for our proposed LISA block.
- In Sec. A4, we provide additional details for D-LISA.
- In Sec. A5, we provide additional qualitative results.

## A1    Additional multi-object grounding results

**F1@0.25 evaluation on Multi3DRefer validation set.** We provide additional comparisons with M3DRef-CLIP over F1@0.25 in Tab. A1. We observe that our D-LISA achieves a better overall F1@0.25 score, especially for multiple targets and sub-categories where the distractors of the same semantic class exist. This aligns with our observation for F1@0.5 results in Tab. 4.

Table A1: F1@0.25 results on the Multi3DRefer validation set.

| Method | F1@0.25 on Val ($\uparrow$) | | | | | |
|---|---|---|---|---|---|---|
| | ZT w/o D | ZT w/D | ST w/o D | ST w/D | MT | All |
| M3DRef-CLIP | 81.8 | 39.4 | 53.5 | 34.6 | 43.6 | 42.8 |
| M3DRef-CLIP w/NMS | 79.0 | 40.5 | **76.9** | 46.8 | 57.0 | 56.3 |
| D-LISA | **82.4** | **43.7** | 75.5 | **49.3** | **58.4** | **57.8** |

**Additional ablation results on question types.** Additional ablations for different query types, including queries with spatial, color, texture, and shape information are reported in Tab. A2. We observe that each proposed module effectively improves the performance for the queries that contain spatial, color, and shape information, and is competitive with the baseline for queries with texture information. The overall model achieves better grounding performance across all query types than the baseline.

Table A2: Ablation studies on question types on Multi3DRefer dataset. 'LIS.', 'DBP.' and 'DMR.' stands for 'Language informed spatial fusion', 'Dynamic box proposal', and 'Dynamic multi-view renderer' respectively. F1@0.5 results are reported.

| Module | | | Question Type | | | |
|---|---|---|---|---|---|---|
| LIS. | DBP. | DMR. | Spatial | Color | Texture | Shape |
| ✗ | ✗ | ✗ | 48.9 | 50.8 | 52.1 | 51.7 |
| ✗ | ✗ | ✓ | 49.4 | 51.1 | 51.7 | 51.8 |
| ✗ | ✓ | ✗ | 49.9 | 51.4 | 51.8 | 53.0 |
| ✓ | ✗ | ✗ | 50.0 | 51.7 | **53.4** | 52.4 |
| ✓ | ✓ | ✓ | **50.9** | **52.1** | 52.9 | **53.3** |

**Additional ablation results on the filtering threshold $\tau_f$.** To determine the optimal filtering threshold $\tau_f$ in Eq. (2), we conduct experiments with different filtering threshold on Multi3DRefer dataset. The result is shown in Tab. A3. We observe that using 0.5 results in the best performance.

Table A3: Ablation studies on the filtering threshold $\tau_f$. F1@0.5 results are reported.

| $\tau_f$ | 0.4 | 0.5 | 0.6 |
|---|---|---|---|
| F1@0.5 ($\uparrow$) | 50.0 | **51.2** | 49.0 |

**Additional comparison on the NMS module.** We show the additional comparison between our proposed D-LISA and D-LISA without NMS on Multi3DRefer in Tab. A4. We observe that our

Table A4: Ablation studies on the NMS module. F1@0.5 results are reported.

| NMS | M3DRef-CLIP | D-LISA |
|---|---|---|
| ✗ | 38.4 | 39.8 |
| ✓ | 49.3 | **51.2** |

designed D-LISA outperforms the baseline M3DRef-CLIP both with and without the NMS module. Using the NMS module would lead to a higher F1 score compared to not using it.

## A2 Additional single-object grounding comparisons

We provide additional comparisons with state-of-the-art methods on ScanRefer and Nr3D datasets for single-object grounding. These methods do not follow the detection-and-selection two-stage diagram. Different from ScanRefer, Nr3D assumes *perfect object proposals are provided*. We focus on the grounding performance on the ScanRefer dataset as the task setting is more realistic. We report the grounding performance on both ScanRefer and Nr3D for completeness.

Table A5: Grounding Acc@0.5 of additional methods on the ScanRefer dataset [8].

| Method | Acc@0.5 on Val (↑) | | | Acc@0.5 on Test (↑) | | |
|---|---|---|---|---|---|---|
| | Unique | Multiple | All | Unique | Multiple | All |
| SAT [46] | 50.8 | 25.2 | 30.1 | - | - | - |
| MVT [21] | 66.5 | 25.3 | 33.3 | - | - | - |
| 3D-SPS [30] | 66.7 | 29.8 | 37.0 | - | - | - |
| ViL3DRef [11] | 68.6 | 30.7 | 37.7 | - | - | - |
| 3DRP-Net [38] | 67.7 | 32.0 | 38.9 | - | - | - |
| BUTD-DETR [24] | 66.3 | 35.1 | 39.8 | - | - | - |
| 3D-VisTA(scratch) [54] | 70.9 | 34.8 | 41.5 | - | - | - |
| ConcreteNet [34] | **75.6** | 36.6 | 43.8 | **69.3** | 37.6 | 44.7 |
| DOrA [39] | - | - | 44.8 | - | - | - |
| D-LISA | 75.5 | **40.0** | **46.9** | 69.0 | **39.7** | **46.3** |

**ScanRefer dataset.** We provide additional comparisons with other state-of-the-art methods on the ScanRefer dataset in Tab. A5. For the methods using object proposals as input instead of the 3D scene, typically a separate pre-trained detector is used to pre-process the scene [11, 21, 39, 46, 54]. Our D-LISA outperforms all existing methods and achieves the best grounding accuracy on both the validation set and test set, which further validates the effectiveness of our proposed modules.

**Nr3D dataset.** We provide additional comparisons with other state-of-the-art methods on the Nr3D dataset in Tab. A6. Our D-LISA still achieves comparable results.

## A3 Additional results for LISA

We provide more experimental results on our designed language informed spatial attention (LISA) module. We show the ablation results on the design choice and compare our module with other language-guided attention modules.

**Design choice.** We analyze the factors that affect the spatial score $\beta$ and report the F1@0.5 metric on Multi3DRefer dataset in Tab. A7. The result shows using both the sentence feature and object feature to predict the spatial score $\beta$ yields the best grounding performance.

**Additional comparison.** We compare our designed LISA with the spatial self-attention in ViL3DRef [11], which also models the object relations guided by language. ViL3DRef pre-defines object relations through hand-crafted features. These hand-selected features work with ground truth object proposals but lead to worse performance when the object proposals are predicted, i.e. noisy. As is shown in Tab. A5 and Tab. A6, though ViL3DRel works well on the Nr3D benchmark which provides ground truth box proposals, the performance is much worse when validating on the ScanRefer benchmark where no ground truth proposals are provided.

Table A6: Grounding accuracy of additional methods on the Nr3D dataset [2].

| Method | Accuracy on Val (↑) | | | | |
|---|---|---|---|---|---|
| | Easy | Hard | View-Dep | View-Indep | All |
| TransRefer3D [18] | 48.5 | 36.0 | 36.5 | 44.9 | 42.1 |
| LanguageRefer [32] | 51.0 | 36.6 | 41.7 | 45.0 | 43.9 |
| LAR [4] | 56.1 | 41.8 | 46.7 | 50.2 | 48.9 |
| SAT [46] | 56.3 | 42.4 | 46.9 | 50.4 | 49.2 |
| 3D-SPS [30] | 58.1 | 45.1 | 48.0 | 53.2 | 51.5 |
| BUTD-DETR [24] | 60.7 | 48.4 | 46.0 | 58.0 | 54.6 |
| MVT [21] | 61.3 | 49.1 | 54.3 | 55.4 | 55.4 |
| 3D-VisTA(scratch) [54] | 65.9 | 49.4 | 53.7 | 59.4 | 57.5 |
| DOrA [39] | 59.7 | 66.6 | 53.1 | 59.2 | 59.9 |
| CoT3DRef [5] | 70.4 | 57.3 | 61.5 | 64.8 | 64.0 |
| ViL3DRef [11] | 70.2 | 57.4 | 62.0 | 64.5 | 64.4 |
| 3DRP-Net [38] | **71.4** | **59.7** | **64.2** | **65.2** | **65.9** |
| D-LISA | 60.2 | 46.2 | 44.3 | 57.4 | 53.1 |

Table A7: Ablation study of different design choices for LISA on Multi3DRefer dataset.

| Sentence | Object | F1@0.5 on Val (↑) | | | | | |
|---|---|---|---|---|---|---|---|
| | | ZT w/o D | ZT w/D | ST w/o D | ST w/D | MT | All |
| ✗ | ✗ | 79.0 | 40.5 | 67.6 | 40.0 | 49.1 | 49.3 |
| ✓ | ✗ | 77.8 | 39.4 | **67.9** | 41.4 | 49.2 | 50.0 |
| ✓ | ✓ | **80.1** | **42.6** | 66.6 | **41.8** | **49.9** | **50.4** |

We substitute LISA with the spatial self-attention in ViL3DRef and report the F1@0.5 metric on Multi3DRefer dataset in Tab. A8. Our proposed LISA achieves better grounding performance with simpler relation representation.

## A4 Additional details for D-LISA

We provide additional architecture details for our D-LISA and additional implementation details for the experiment setup.

**Cross-attention.** In the language informed fusion module, a language informed spatial attention block is followed by a cross-attention block (Sec. 3.2). The cross-attention block takes the spatially enhanced visual features $\boldsymbol{F}^s$ from LISA and word features after a self-attention block as input and generates language-informed visual features $\boldsymbol{F}^c$. We follow the standard cross-attention mechanism as described in Vaswani et al. [35]. We formulate the word feature matrix input as $\boldsymbol{F}_T = [\boldsymbol{t}_1, \boldsymbol{t}_2, \ldots, \boldsymbol{t}_L]^T \in \mathbb{R}^{d \times d}$, where $\boldsymbol{t}_j \in \mathbb{R}^d$ is the corresponding feature for $\boldsymbol{w}_j \in \mathcal{W}$ after self-attention. Given $\boldsymbol{F}^s$ and $\boldsymbol{F}_T$, queries $\boldsymbol{Q}_c$, keys $\boldsymbol{K}_c$ and values $\boldsymbol{V}_c$ correspond to:

$$\boldsymbol{Q}_c = \boldsymbol{F}^s \boldsymbol{W}_Q^c, \quad \boldsymbol{K}_c = \boldsymbol{F}_T \boldsymbol{W}_K^c, \quad \boldsymbol{V}_c = \boldsymbol{F}_T \boldsymbol{W}_V^c \tag{A13}$$

with linear projections $\boldsymbol{W}_{Q/V/K}^c \in \mathbb{R}^{d \times d}$. The overall cross-attention is formulated as:

$$\boldsymbol{F}^c = \text{Cross-Attention}(\boldsymbol{F}^s, \boldsymbol{F}_T) = \text{softmax}(\frac{\boldsymbol{Q}_c \boldsymbol{K}_c^T}{\sqrt{d}})\boldsymbol{V}_c, \tag{A14}$$

where $\text{softmax}$ is the softmax normalization along rows.

**Additional implementation details.** Following the prior work [52], we take point coordinates, point normals, and per-point multi-view features $\mathcal{S} \in \mathbb{R}^{H \times (3+3+128)}$ as scene input, where $H$ denotes the total number of points in the scene. For NMS process, we set the threshold $\tau_{\text{NMS}}$ to be 0.4. For CLIP, we use a frozen pre-trained CLIP with ViT-B/32. For loss coefficient terms in Eq. (12), we set $\lambda_{\text{det}}$, $\lambda_{\text{ref}}$ and $\lambda_{\text{ctr}}$ to 1 and $\lambda_{\text{dyn}}$ to 5. We initialize the camera baseline poses following the fixed camera poses in prior work [52], where for each view the rendering camera is set to be 1 meter away from the object, with an elevation angle of $45°$. For the fusion module, we follow the same settings in terms of dimension size, layer number, and head size as used for the baseline [52].

Table A8: Comparison of language guided spatial attention methods on Multi3DRefer dataset.

| Attention module | F1@0.5 on Val (↑) | | | | | |
|---|---|---|---|---|---|---|
| | ZT w/o D | ZT w/D | ST w/o D | ST w/D | MT | All |
| Spatial Self-Attention [11] | 79.0 | 31.2 | **66.6** | 39.8 | 49.0 | 48.7 |
| LISA | **80.1** | **42.6** | **66.6** | **41.8** | **49.9** | **50.4** |

Figure A1: **Additional qualitative examples of Multi3DRefer val set in MT category.** For each scene-text pair, we visualize the predictions of M3DRef-CLIP, M3DRef-CLIP w/NMS, D-LISA and ground truth labels in magenta/blue/green/red separately.

## A5   Additional qualitative results

We provide additional qualitative comparisons for MT category (Fig. A1) and ST w/D category (Fig. A2). For MT category examples, our D-LISA successfully identifies all objects that match the text description. For ST w/D category examples, our D-LISA accurately identifies the object without being distracted by the distractors.

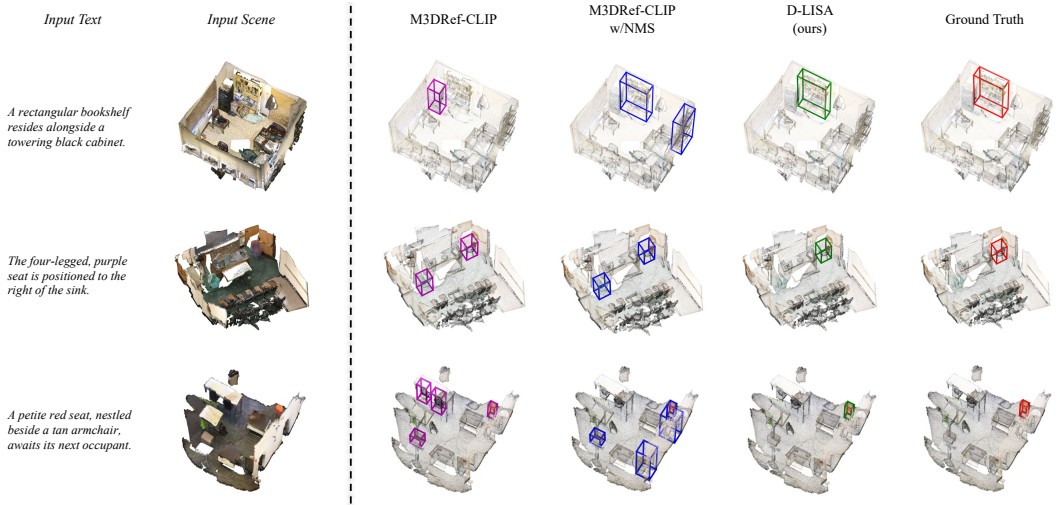

Figure A2: **Additional qualitative examples of Multi3DRefer val set in ST w/D category.** For each scene-text pair, we visualize the predictions of M3DRef-CLIP, M3DRef-CLIP w/NMS, D-LISA and ground truth labels in magenta/blue/green/red separately.

