# OpenReview forum: "Multi-Object 3D Grounding with Dynamic Modules and Language-Informed Spatial Attention"
_NeurIPS.cc/2024/Conference — NeurIPS 2024 poster_

### Official Review · Reviewer_rjTU · 2024-07-05

**Soundness:** 2
**Presentation:** 2
**Contribution:** 3
**Rating:** 4
**Confidence:** 4

**Summary:**

This paper improves upon the previous work, M3DRef-CLIP, through three key modifications: First, the authors incorporate an additional proposal probability prediction branch followed by a NMS operator to filter out low-confidence and redundant object proposals. Second, they learn camera pose residuals to dynamically render the object proposals from multiple views and extract multi-view CLIP features as the 2D feature. Third, they incorporate spatial relations weighted by learnable weights from visual and text features. The proposed method outperforms M3DRef-CLIP in several settings and datasets.

**Strengths:**

•  The proposed method demonstrates state-of-the-art performance in both multi-object and single-object 3D grounding tasks.

•  The ablation study clearly shows the performance improvement brought by each modification.

**Weaknesses:**

1. Despite the performance improvement, the time and memory complexity of the proposed methods seem substantial, particularly the multi-view rendering step. It would be beneficial to include a complexity comparison with baseline methods.

2. In M3DRef-CLIP, both F1@0.25 and F1@0.5 are reported. Including the same evaluation metrics would be advantageous, as this paper is an improvement over M3DRef-CLIP.

3. The paper would benefit from more detailed motivations and explanations. For example, what motivates the design of camera pose offset prediction (Eq. 5) in this manner? Why is the designed LISA module better than the language-conditioned spatial self-attention in ViL3DRel [9]?

4. What would the performance be if the camera pose offset were removed while keeping the average of multi-view CLIP features?

5.  Is there any supervision on the predicted proposal probability (Eq. 1)? How does the threshold in Eq. 2 affect the performance?

**Questions:**

Please refer to the questions in the weaknesses section. I would consider increasing the rating if the authors can address my questions and concerns properly.

**Limitations:**

The authors have discussed some limitations and indicated that there are no potential societal impacts of the proposed method.

---

> ### Author Rebuttal · Authors · 2024-08-06
>
> ## To Reviewer rjTU
>
> ```
> Q10. Time and memory complexity comparisons
> ```
>
> Thanks for the suggestion. We show the FLOPs and inference time of each proposed module and a comparison with the baseline model M3DRef-CLIP in Tab. R5. All experiments are conducted on Multi3DRefer validation set on a single NVIDIA A100 GPU. The reported FLOPs and inference time are the averaged over the validation set.
>
>
> We observe that the dynamic box proposal module and the dynamic multi-view renderer in the dynamic vision module contribute marginally to the computation. The additional computation in the language-informed spatial fusion module is also minimal. In other words, our model achieves better grounding performance without significantly increasing computations.
>
> **Table R5. Computational cost for proposed modules**
> | Module                       | FLOPs    | Inference time |
> |:-|:-:|:-:|
> | Baseline detector            | 943.1 M  | 0.235 s        |
> | Detector w/ dynamic box proposal | 943.1 M  | 0.241 s        |
> | Baseline renderer            | 638.9 G  | 0.271 s        |
> | Dynamic multi-view renderer  | 638.9 G  | 0.276 s        |
> | Baseline fusion module       | 155.3 M  | 0.004 s        |
> | Language-informed spatial fusion | 247.4 M  | 0.007 s        |
>
> ```
> Q11. Comparison of the additional metric F1@0.25 with M3DRef-CLIP
> ```
>
> Please note, M3DRef-CLIP did not provide quantitative results over F1@0.25 in their tables. Here, we use their model to provide additional comparisons with M3DRef-CLIP over F1@0.25 in Tab. R6. We observe that our D-LISA achieves a better overall F1@0.25 score, especially for multiple targets and sub-categories where the distractors of the same semantic class exist. This aligns with our observation for F1@0.5 results in Tab 4.
>
> **Table R6. F1@0.25 (&uparrow;) result on Multi3DRefer validation set**
> | Model           | ZT w/o D | ZT w/D | ST w/o D | ST w/D | MT   | All  |
> |:-|:-:|:-:|:-:|:-:|:-:|:-:|
> | M3DRef-CLIP     | 81.8     | 39.4   | 53.5     | 34.6   | 43.6 | 42.8 |
> | M3DRef-CLIP w/NMS | 79.0   | 40.5   | **76.9**     | 46.8   | 57.0 | 56.3 |
> | D-LISA          | **82.4**     | **43.7**   | 75.5     | **49.3**   | **58.4** | **57.8** |
>
> ```
> Q12. What is the motivation for camera pose offset prediction?
> ```
>
> In L30-32, we motivated that the viewpoints of the renderer may need to be different across different scenes and object sizes. To avoid poor viewpoint initializations, we start from the fixed viewpoints and predict an offset for each viewpoint. In Eq. 5, we use an MLP to learn the camera pose offset for each view based on the average box size, as the object size is a strong indicator when choosing the rendering camera pose.
>
> ```
> Q13. Why is the proposed LISA better than the language-conditioned spatial self-attention in ViL3DRel?
> ```
>
> ViL3DRef [C] incorporates hand-selected features to guide spatial relations. While these hand-selected features perform well when provided with perfect (ground-truth) object proposals, they are not robust to errors, a typical issue with hand-designed features. In particular, when these features are used with detected (noisy) object proposals, their performance decreases.
>
> As is shown in Tab. A1 and Tab. A2, though ViL3DRel works well on the Nr3D benchmark which provides ground truth box proposals, the performance is much worse when validating on the ScanRefer benchmark where no ground truth proposals are provided. We compare our LISA and the language-conditioned spatial self-attention in ViL3DRel on the Multi3DRefer benchmark in Tab. A4. The results validate the effectiveness of our module.
>
> * [C] S. Chen, P.-L. Guhur, M. Tapaswi, C. Schmid, and I. Laptev. "Language conditioned spatial relation reasoning for 3d object grounding". In NeurIPS, 2022
>
> ```
> Q14. What would the performance be if camera pose offset were removed?
> ```
>
> Please see the ablation study in rows 1 and 2 of Tab. 4 where using the camera pose offset (DMR) improves the model performance.
>
> ```
> Q15. How is the predicted proposal probability supervised?
> ```
>
> The predicted probability is end-to-end supervised by weighting the detector features with the probabilities, as is shown in Eq. 4. To encourage fewer numbers of boxes, we also propose to regularize with the loss proposed in Eq. 3.
>
> ```
> Q16. How does the threshold 0.5 in Eq.2 affect the performance?
> ```
>
> Thanks for the suggestion. As we used a Sigmoid nonlinearity in Eq. 1, the maximum-likelihood estimate results in a threshold of 0.5. Here in Tab. R7, we further conduct experiments, following the setting in Sec. 4.3, but using different thresholds. We observe that using 0.5 results in the best performance.
>
> **Table R7. Additional ablation studies on the filtering threshold.**
> | Threshold | 0.4  | 0.5  | 0.6  |
> |:-|:-:|:-:|:-:|
> | F1@0.5 (&uparrow;)   | 49.2 | **50.3** | 47.2 |

---

> > ### Comment · Reviewer_rjTU · 2024-08-12
> > **Response to rebuttal**
> >
> > Thank you to the authors for providing the rebuttal. However, most of my concerns remain inadequately addressed. Here are some additional comments:
> >
> > 1. I am puzzled by the complexity comparison between the baseline renderer and the dynamic multi-view renderer. How is it that the multi-view renderer and CLIP feature extraction have almost no impact on memory and time?
> >
> > 2. The LISA method also computes the pairwise L2 distance between box centers, which is a feature in Vil3DRef. Shouldn't this distance calculation also be affected by noisy detected proposals?
> >
> > 3. Rows 1 and 2 of Table 4 do not show the performance when removing the camera pose offset while keeping the average of multi-view CLIP features.
> >
> > 4. The results in Table R7 do not match the performance reported in Table 4, which is quite confusing.
> >
> > 5. I agree with Reviewer su8L and WVQA that the three improvements over M3DRef-CLIP proposed in this paper are more engineering-focused and show limited novelty.
> >
> > Hence, I would like to maintain my initial rating.

---

> > > ### Author Response · Authors · 2024-08-12
> > >
> > > Thanks for the careful review and additional feedback. We address the additional concerns below:
> > >
> > > ```
> > > 1. Complexity of the dynamic multi-view renderer.
> > > ```
> > > Sorry for the confusion. Recall that the baseline renderer in M3DRef-CLIP **also needs to render from multi-views** and **extracts the CLIP features**. The additional overhead brought by our dynamic multi-view renderer is **only computing the camera viewpoint offset prediction**, which has little impact on the overall computation.
> > >
> > > ```
> > > 2. Would L2 distance also be affected by noisy detected proposals?
> > > ```
> > > Yes, we agree that L2 distance would be affected by noisy detected proposals. However, prior works, like M3DRef-CLIP and 3DVG-Transformer [D], have demonstrated the effectiveness of L2 distance in modeling spatial relationships in the noisy detected proposals setting. Our method do not rely on other hand-crafted features.
> > >
> > > * [D] L. Zhao, D. Cai, L. Sheng, and D. Xu. 3DVG-Transformer: Relation modeling for visual grounding on point clouds. In ICCV, 2021.
> > >
> > > ```
> > > 3. What would the performance be if camera pose offset were removed? (Q14)
> > > ```
> > > Again, M3DRef-CLIP's baseline renderer **also  renders from multi-views** and **extracts the CLIP features**. So the ablation study in row 1 of Tab. 4 is exactly the performance when the camera pose offsets were removed.
> > >
> > > ```
> > > 4. The results in Table R7 do not match the performance reported in Table 4.
> > > ```
> > > Sorry for the confusion. Due to limited rebuttal time and computational resources, we conduct the ablation on the dynamic proposal module only. The experiment follows the setting in row 3 of Tab. 4. We will provide an ablation on the filtering threshold for the full model.
> > >
> > > ```
> > > 5.  Novelty and Engineering-focused of the proposed modules
> > > ```
> > > Thanks for the clarification. We believe our approach is sufficiently novel and has not been done by prior works. Next, we do not share the sentiment that  "engineering-focused" correlates with a lack of novelty. Our proposed modules are well-motivated and lead to an effective system that outperforms SOTA.

---

### Official Review · Reviewer_WVQA · 2024-07-08

**Soundness:** 3
**Presentation:** 3
**Contribution:** 2
**Rating:** 3
**Confidence:** 5

**Summary:**

This paper introduces a novel two-stage approach for multi-object 3D grounding from a point cloud based on a given query phrase. The first stage of D-LISA uses a dynamic proposal module that selects a variable number of box proposals instead of a fixed maximum, addressing the issue of determining the optimal number of proposals in the scene. D-LISA incorporates a dynamic multi-view renderer module that optimizes viewing angles for each proposal based on the specific scene, moving away from the fixed camera poses used in prior work. The second stage introduces a module that reasons over the spatial relationships among objects, guided by the textual description, improving the contextual understanding of the model. Experiments conducted on the Multi3DRefer benchmark demonstrate that D-LISA outperforms the state-of-the-art methods by a significant 12.8% absolute increase in multi-object 3D grounding performance. It also shows competitive results in single-object 3D grounding tasks.

**Strengths:**

1. The method shows a substantial improvement over existing baselines, indicating effective handling of complex 3D scenes with multiple objects.
2. The dynamic module can reduce the proposals effectively.

**Weaknesses:**

1. The novelty of the dynamic vision module is limited. In fact, I think the statement of dynamic vision module is kind of exaggeration. From my perspective, the authors just calculate the probability of each box candidate to remove the low-probability boxes and use NMS to filter overlapping boxes. Both the two operations has nothing to do with a novel dynamic vision module.
2. Also, the novelty of the LISA module is also doubtful. The authors use spatial scores to modulate the self-attention and spatial distance matrix. Although this operation is innovative to some extent, it is not enough for acceptance. It is more like a trick rather than a novelty.

**Questions:**

Could the authors re-organize the novelty according to the weakness to highlight the key points more clearly?

**Limitations:**

Limitations are adequately addressed.

---

> ### Author Rebuttal · Authors · 2024-08-06
>
> ## To Reviewer WVQA
>
> ```
> Q8. The dynamic vision module only removes the low-probability boxes and uses NMS to filter overlapping boxes, which is not novel.
> ```
>
> We believe the reviewer is referring to the dynamic box proposal module. For the dynamic box proposal module, we do not claim NMS to be the novel contribution. We clearly state *"we employ a dynamic box proposal module **with** non-maximum suppression (NMS)"* in L98-99.
>
> Our contribution to this module is the learning of dynamic proposal probability, supervised by the new dynamic proposal loss introduced in L113-116. This enables the end-to-end training by weighting the detector features with the probabilities, as is shown in Eq. 4. Additionally, we also propose the dynamic multi-view renderer to have dynamic camera viewpoints based on different scenes.
>
> We respectfully disagree with the reviewer that the proposed method is not novel. To the best of our knowledge, prior works in 3D grounding have not studied these aspects. We would be grateful if the reviewer could provide references to back up this novelty claim.
>
> ```
> Q9. Novelty of the LISA module
> ```
>
> Our baseline model M3DRef-CLIP follows 3DVG-Transformer, which directly uses spatial distances as additional attention weights without further reasoning. Prior works, like ViL3DRef [A] and CORE-3DVG [B] explored the spatial relations with hand-selected features. These hand-selected features work with ground truth object proposals but lead to worse performance when the object proposals are predicted, i.e. noisy. As is shown in Tab. A1 and Tab. A2, though ViL3DRel works well on the Nr3D benchmark which provides ground truth box proposals, the performance is much worse when validating on the ScanRefer benchmark where no ground truth proposals are provided.
>
> To address these shortcomings, we propose to use language-guided spatial scores to balance the visual attention weights and spatial attention weights. We believe our LISA design greatly differs from the existing works.
>
> * [A] S. Chen, P.-L. Guhur, M. Tapaswi, C. Schmid, and I. Laptev. "Language conditioned spatial relation reasoning for 3d object grounding". In NeurIPS, 2022
> * [B] L. Yang, Z. Zhang, Z. Qi, Y. Xu, W. Liu, Y. Shan, B. Li, W. Yang, P. Li, Y. Wang, et al. "Exploiting contextual objects and relations for 3d visual grounding". In NeurIPS, 2023.

---

### Official Review · Reviewer_W21s · 2024-07-10

**Soundness:** 3
**Presentation:** 4
**Contribution:** 2
**Rating:** 5
**Confidence:** 4

**Summary:**

This paper proposes D-LISA, a two-stage framework for multi-object 3D grounding. D-LISA consists of three novel components that make the method effective, namely a dynamic box proposal module, a dynamic multi-view renderer and a language informed spatial fusion module. Comprehensive Experiments are done on Multi3DRefer, ScanRefer and Nr3D datasets to prove the superiority of D-LISA over previous methods. Experimental results show that D-LISA not only outperforms previous methods on multi-object grounding, but also achieves comparable results on single-object grounding.

**Strengths:**

1. The paper is well-written and easy to understand and the figures help with the illustration of the overall idea.
2. Comprehensive experiments and ablation studies are conducted.
3. Implementation and evaluation details are clearly provided, making this work easy to follow.

**Weaknesses:**

1. In Table A2, the proposed method falls behind state-of-the-art method by a large margin, which somehow is not competitive.
2. In the ablation study (Table 4), I'm curious what would happen if putting two modules together, e.g. abandon LIS and use DBP and DMR, since the improvements of the full model is not significant compared to the model with one component.
3. The analysis of dynamic pose distribution is lacking. Readers can't tell from Figure 4 (b) if the dynamic pose rendered results are better. Fixed pose results seem to be looking at the object from a more informative view.
4. How much effect NMS has on the proposed method seems to be unclear, the paper only conducted experiments by adding NMS to a baseline method, resulting in a much better performance.

**Questions:**

Please see the weakness part.

**Limitations:**

The authors have adequately addressed the limitations and potential negative societal impact of their work.

---

> ### Author Rebuttal · Authors · 2024-08-06
>
> ## To Reviewer W21s
>
> ```
> Q4. Comparisons with SOTA methods on Nr3D benchmark in Tab. A2.
> ```
>
> As is mentioned in L415-416, the Nr3D benchmark **assumes perfect object proposals**, which is not the most realistic setting. Hence, we follow M3DRef-CLIP to consider the setting where object proposals need to be detected.
>
> We note that M3DRef-CLIP has made a similar observation: models designed for non-perfect objective proposals generally perform less effectively in the perfect object proposal setting. In Tab. A2, we report the comparisons for completeness of the evaluation. However, we do not believe this is a fair comparison as the models are designed with different intentions.
>
> ```
> Q5. Additional ablation study on the proposed modules.
> ```
>
> Here we provide additional ablation by gradually adding proposed modules in Tab. R3. The experiments follow the setting in Tab. 4. We observe that each proposed module improves the model. The overall model with all proposed modules achieves the best result.
>
> **Table R3. Additional ablation studies on combined modules.**
> | LIS      | DBP       | DMR       | F1@0.5(&uparrow;) |
> |:-:|:-:|:-:|:-:|
> | &cross;   | &cross;    | &cross;    | 49.3   |
> | &check;   | &cross;    | &cross;    | 50.4   |
> | &check;   | &check;    | &cross;    | 50.9   |
> | &check;   | &check;    |  &check;   | **51.2**   |
>
> ```
> Q6. Why is the rendering with dynamic pose better?
> ```
>
> Sorry for the confusion. It is difficult to judge what is *"more informative"* from a model perspective. The provided illustrations in Fig. 4 (a) and (b) meant to show the differences in the learned poses versus the fixed pose. Empirically, we observe that the dynamic camera pose leads to better grounding performance, see row 1 and row 2 in Tab. 4.
>
> ```
> Q7. What is the effect of NMS on the proposed method?
> ```
>
> As is mentioned in L214-216, adding an NMS module removes the duplicate predictions and leads to a higher recall, thus improving the F1 score. The state-of-the-art method M3DRef-CLIP did not include the NMS module in their pipeline. We show the effect of the NMS module by adding a baseline of M3DRef-CLIP with NMS in Tab. 1.
>
> To provide a more comprehensive analysis of the NMS module, we show the additional comparison between our proposed D-LISA and D-LISA **without** NMS on Multi3DRefer. See Tab. R4 below.  We observe that our designed D-LISA outperforms the baseline M3DRef-CLIP both with and without the NMS module. Using the NMS module would lead to a higher F1 score compared to not using it.
>
> **Table R4. The F1@0.5 (&uparrow;) w/ and w/o the NMS module.**
> |NMS| M3DRef-CLIP | D-LISA |
> |:-:|:-:|:-:|
> | &cross; | 38.4   | 39.8   |
> | &check; | 49.3   | **51.2**   |

---

> > ### Comment · Reviewer_W21s · 2024-08-12
> >
> > Thank you for the thorough experiments and ablations. They addressed the concerns I had. The work has been made more comprehensive with the experimental results provided. However, I agree with the other reviewers that the work is introducing useful techniques for multi-object grounding from an engineering perspective, therefore I'd like to maintain my score.

---

> > > ### Author Response · Authors · 2024-08-13
> > >
> > > Dear Reviewer W21s,
> > >
> > > Thank you for the feedback. We are glad that we could address your concerns.
> > >
> > > We believe that our approach is sufficiently novel and that improving the engineering perspective of a method to build an effective system is important. However, we understand and respect the difference in opinion.
> > >
> > > Best,
> > >
> > > Authors

---

### Official Review · Reviewer_su8L · 2024-07-20

**Soundness:** 3
**Presentation:** 2
**Contribution:** 3
**Rating:** 5
**Confidence:** 4

**Summary:**

The paper introduces D-LISA, a two-stage approach for multi-object 3D grounding that incorporates three innovative modules. First, a dynamic vision module generates variable and learnable box proposals. Second, a dynamic multi-view renderer extracts features from optimized viewing angles. Third, a language-informed spatial attention module reasons over the proposals to output final predictions. Empirically, D-LISA outperforms state-of-the-art methods by 12.8% in multi-object 3D grounding and is competitive in single-object 3D grounding.

**Strengths:**

1. This paper is generally well-written and clearly stated.
2. The key idea lies in enhancing visual understanding and human-computer interaction by improving the ability to locate objects in 3D scenes based on natural language descriptions.
3. Experiments demonstrate that D-LISA outperforms the existing state-of-the-art, indicating the effectiveness of the proposed innovations.

**Weaknesses:**

1. It is not clear what core issue this paper is targeting in the task of dynamic multi-object 3D grounding. I suggest the authors include this in the abstract and introduction section. To me, the three improvements over M3DRef-CLIP are very engineering.
2. The paper could benefit from a more detailed ablation study that isolates the impact of each dynamic component (the dynamic proposal module, the dynamic multi-view renderer, and the LISA module) on different types of scenes and queries to better understand their individual contributions.
3. The paper does not provide details on the computational cost of the dynamic components, such as the multi-view rendering and the language-informed spatial attention module.

**Questions:**

See weakness.

**Limitations:**

Yes.

---

> ### Author Rebuttal · Authors · 2024-08-06
>
> ## To Reviewer su8L
> ```
> Q1. Core issues targeted for multi-object 3D grounding
> ```
>
> In L26-33, we summarized the targeted issues and the proposed solutions. Concretely, we identified that:
> - object proposals are selected based on a **fixed** maximum number,
> - the feature extractions from the proposals are based on a set of **fixed** camera poses for all scenes,
> - the fusion module, of the existing method, lacks effective reasoning over spatial relations among objects.
>
> We believe these aspects are crucial to building an effective multi-object 3D grounding system. We then propose a dynamic vision module and a language-informed spatial fusion module to address these issues. We will clarify the introduction.
>
> ```
> Q2. Ablation studies on different types of scenes and queries
> ```
>
> We evaluate the contribution of each proposed component in Tab. 4. The performance of individual modules is shown in rows 2, 3, and 4 respectively. The different categories of scenes and queries are introduced in L197-201, including
> - zero target without distractors of the same semantic class (ZT w/o D);
> - zero target with distractors (ZT w/D);
> - single target without distractors (ST w/o D);
> - Single target with distractors (ST w/D);
> - multiple targets (MT).
>
> These scene and query categories follow the prior work M3DRef-CLIP.
>
> Additional ablations for different query types, including queries with spatial, color, texture, and shape information are reported in the table below. In Tab. R1, We report the F1@0.5 (&uparrow;) metric on the Multi3DRefer validation set. We observe that each proposed module effectively improves the performance for the queries that contain spatial, color, and shape information, and is competitive with the baseline for queries with texture information. The overall model achieves better grounding performance across all query types than the baseline.
>
> **Table R1. Additional ablation studies on question types**
> | LIS       | DBP      | DMR       | Spatial  | Color  | Texture | Shape |
> |:-:|:-:|:-:|:-:|:-:|:-:|:-:|
> | &cross;    | &cross;    | &cross;    |  48.9    | 50.8   | 52.1    | 51.7  |
> | &cross;    | &cross;    | &check;    |  49.4    | 51.1   | 51.7    | 51.8  |
> | &cross;    | &check;    | &cross;    |  49.9    | 51.4   | 51.8    | 53.0  |
> | &check;    | &cross;    | &cross;    |  50.0    | 51.7   |**53.4** | 52.4  |
> | &check;    | &check;    | &check;    | **50.9** |**52.1**| 52.9    | **53.3** |
>
> ```
> Q3. The computational cost for each proposed module
> ```
>
> Thanks for the suggestion. We show the FLOPs and inference time of each proposed module and a comparison with the baseline model M3DRef-CLIP in Tab. R2. All experiments are conducted on Multi3DRefer validation set on a single NVIDIA A100 GPU. The reported FLOPs and inference time are averaged over the validation set.
>
> We observe that the dynamic box proposal module and the dynamic multi-view renderer in the dynamic vision module contribute marginally to the computation. The additional computation in the language-informed spatial fusion module is also minimal. In other words, our model achieves better grounding performance without significantly increasing computations.
>
> **Table R2. Computational cost for proposed modules**
> | Module                    | FLOPs    | Inference time |
> |:-|:-:|:-:|
> | Baseline detector            | 943.1 M  | 0.235 s        |
> | Detector w/ dynamic box proposal | 943.1 M  | 0.241 s        |
> | Baseline renderer            | 638.9 G  | 0.271 s        |
> | Dynamic multi-view renderer  | 638.9 G  | 0.276 s        |
> | Baseline fusion module       | 155.3 M  | 0.004 s        |
> | Language-informed spatial fusion | 247.4 M  | 0.007 s        |

---

### Author Rebuttal · Authors · 2024-08-06

We thank all the reviewers and the AC for the thorough reviews. We are happy to see the reviewers' supportive comments and feedback. Reviewers **#su8L** and **#W21s** commend the paper for its clear and well-structured writing. Reviewers **#W21s** and **#rjTU** appreciate the comprehensive experiments and ablation studies, with detailed implementation and evaluation. Reviewers **#su8L**, **#WVQA**, and **#rjTU** all acknowledge the substantial improvement in both multi-object and single-object 3D grounding, indicating the effectiveness of the proposed model. We begin the response by restating our contribution, individual questions are addressed in the responses below.

This paper tackles the problem of multi-object 3D grounding where we identify shortcomings in the prior work, e.g., using a fixed number of boxes and features used for reasoning. To address these shortcomings, we propose a dynamic vision module and a language-informed spatial fusion module. For the dynamic vision module, we introduce a dynamic box proposal module that automatically learns the relevant box proposals instead of using a fixed maximum number of proposals. We propose to learn the camera pose for rendering dynamically based on the scene instead of using fixed camera viewpoints. For the language-informed spatial fusion module, we enable efficient reasoning over spatial relations by learning to balance the visual attention weights and spatial relations guided by language. Extensive experiments on both multi-object and single-object 3D grounding benchmarks validate the effectiveness of the proposed model.

We look forward to a constructive discussion period and hope to address any concerns that the reviewers may have!

---

> ### Comment · Area_Chair_kyMS · 2024-08-12
>
> Dear authors, thank you for your rebuttal and response.
>
> I have looked over the paper and reviews and have a small suggestion.  I found the paper title to be somewhat confusing as "Dynamic" is typically refers to static vs dynamic scenes.  I would recommend you consider renaming the paper to
> - "Multi-Object 3D Grounding with Dynamic Proposals and Language Informed Spatial Attention" or
> - "Multi-Object 3D Grounding with Dynamic Modules and Language Informed Spatial Attention"

---

> > ### Author Response · Authors · 2024-08-12
> >
> > Dear AC kyMS,
> >
> > Apologies for the confusion. The word "Dynamic" in the title is intended to describe our approach. We agree that the title could be mistakenly interpreted as referring to a "Dynamic grounding" task.
> >
> > We greatly appreciate the title recommendations. We will be happy to rename the title as suggested!
> >
> > Thank you,
> >
> > Authors

---

### Decision · Program_Chairs · 2024-09-25

**Decision:**

Accept (poster)

**Comment:**

The paper proposes to improve multi-object 3D grounding by introducing a dynamic vision module that selects a variable number of detection based on a proposal probability, camera poses that are predicted by a MLP based on the bounding box size, and a language informed spatial attention.  Experiments on Multi3DRefer and ScanRefer show that the proposed method can performs well.

The submission received divergent ratings with two reviewers (su8L, W21s) giving it a borderline accept and two reviewers (WVQA, rjTU) favoring reject.

Overall, reviewers find the paper to be well-written (su8L,W21s) with experiments that shows the proposed method is effective (su8L,WVQA,rjTU).  However, reviewers had questions about the complexity of the proposed method (especially as it required rendering), and requested additional ablations and clarifications.

While the rebuttal adequately addressed reviewer concerns, reviewers WVQA and rjTU remain negative, stating mainly that the novelty of the proposed modules is limited.  The AC note that WVQA did not follow up during the author response period.

Based on the above, the AC believe that the techniques proposed by the paper are effective (providing a boost in performance over prior work), but there is lack of excitement from reviewers as the techniques are perceived to be more "engineering" and not sufficient "novel".  As novelty is subjective, and the AC believe the authors had done a good job in responding to reviewer questions, the AC feels the paper can be accepted at NeurIPS and to leave it to the community to determine the potential impact of the proposed techniques.

The AC recommend the authors include the additional ablations done for the rebuttal and to update the title of the paper to be clearer.  The paper itself may have also overused the word "dynamic".